# RAP: Risk-Aware Prediction for Robust Planning

Haruki Nishimura*       Jean Mercat*       Blake Wulfe       Rowan McAllister

**Adrien Gaidon**
Toyota Research Institute, USA
`firstname.lastname@tri.global`

**Abstract:** Robust planning in interactive scenarios requires predicting the uncertain future to make risk-aware decisions. Unfortunately, due to long-tail safety-critical events, the risk is often under-estimated by finite-sampling approximations of probabilistic motion forecasts. This can lead to overconfident and unsafe robot behavior, even with robust planners. Instead of assuming full prediction coverage that robust planners require, we propose to make prediction itself risk-aware. We introduce a new prediction objective to learn a risk-biased distribution over trajectories, so that risk evaluation simplifies to an expected cost estimation under this biased distribution. This reduces the sample complexity of the risk estimation during online planning, which is needed for safe real-time performance. Evaluation results in a didactic simulation environment and on a real-world dataset demonstrate the effectiveness of our approach. The code[2] and a demo[3] are available.

**Keywords:** Risk Measures, Forecasting, Safety, Human-Robot Interaction

## 1 Introduction

In safety-critical and interactive control tasks such as autonomous driving, the robot must successfully account for uncertainty of the future motion of surrounding humans. To achieve this, many contemporary approaches decompose the decision-making pipeline into prediction and planning modules [1–5] for maintainability, debuggability, and interpretability. A prediction module, often learned from data, first produces likely future trajectories of surrounding agents, which are then consumed by a planning module for computing safe robot actions. Recent works [6, 7] further propose to couple prediction with risk-sensitive planning for enhanced safety, wherein the planner computes and minimizes a risk measure [8] of its planned trajectory based on probabilistic forecasts of human motion from the data-driven predictor. A risk measure is a functional that maps a cost distribution to a deterministic real number, which lies between the expected cost and the worst-case cost [9].

Although combining data-driven forecasting and risk-sensitive planning has been shown to be effective, there exist several limitations to this approach. First, accurate risk evaluation of candidate robot plans remains challenging, due to inaccurate characterization of uncertainty in human behavior [10] and finite-sampling from the predictor. Some existing methods that promote diversity of prediction (e.g., [11, 12]) may alleviate this issue, but they are not explicitly designed for reliable risk estimation needed for robust planning. Second, endowing an existing planner with risk-sensitivity often requires non-trivial modifications to its internal optimization algorithm [13–15]. This modification can be problematic, if, for example, an autonomy stack already has a dedicated and complex (risk-neutral) planner in use and cannot easily modify its internal optimization algorithms.

To address the above limitations, we propose to consider risk within the *predictor* rather than in the planner. We present a risk-biased trajectory forecasting framework, which provides a general approach to making a generative trajectory forecasting model risk-aware. Our novel method augments a pre-trained generative model with an additional encoding process. This modification changes the

---

*The first two authors contributed equally to this work.

[2] `https://github.com/TRI-ML/RAP`

[3] `https://huggingface.co/spaces/TRI-ML/risk_biased_prediction`

6th Conference on Robot Learning (CoRL 2022), Auckland, New Zealand.

output of the prediction so that it purposefully and deliberately over-estimates the probability of dangerous trajectories. This "pessimistic" forecasting model gives *distributional robustness* (e.g., [16]) to the planner against potential inaccuracies of the human behavior model.

We achieve the pessimistic risk-biased distribution using a novel prediction loss. This shifts the computational burden of drawing many prediction samples that capture rare events from online deployment to offline prediction training. The planner can still obtain an accurate estimate of the risk measure in real-time during deployment with fewer prediction samples required from the biased distribution. Furthermore, our approach also eliminates the need for modifications to the planner's optimization algorithm. Thus, one can achieve enhanced safety by simply replacing a conventional probabilistic motion forecaster with the proposed risk-biased model, while still using the same existing risk-neutral planner. This capability is intended for use in robotic applications where misestimation of risk could lead to injury, including autonomous vehicles and home robots that must operate safely in close proximity to humans.

Specifically, our contributions in this work are as follows:

- We propose a risk-biased trajectory forecasting framework, which makes forecasts more useful for the downstream task and leads to plans that are robust to distribution shifts.
- Our risk-biased model off-loads the heavy computation of risk estimation from online planning, providing risk-awareness to a generic risk-neutral planner.
- We extensively evaluate our proposed approach in simulation with a planner in the loop and offline with complex real-world data.

## 2 Related Work

**Trajectory forecasting from data.** Early trajectory forecasting approaches defined hand-crafted dynamics models [17, 18], and incorporated rules that induce obstacle avoidance behavior [19] or mimic the overall traffic flow [20, 21]. More recently, data-driven, learning-based methods have gained popularity for their ability to better capture the complexity of human behavior [22], and typically use neural networks defining multi-modal trajectory distributions [12, 23–38].

Significant effort is directed toward increasing the coverage, or diversity, of motion forecasting models [11, 12, 33–41] in order to ensure that no critical events are missed. Diversity can be explicitly encouraged using a best-of-many loss [25], by replacing a mean-squared loss with a Huber loss [40], by choosing trajectory samples that maximize the distribution coverage [34], or by setting diverse anchors or target points [36–38]. Another strategy to increase mode coverage takes advantage of the latent distribution of CVAEs [5, 11, 41] or GANs [12]. Cui et al. [5] argue that besides coverage, sample efficiency is also an important factor. The authors trained a road-scene motion forecasting model to produce predictions of other agents that induce diverse reactions from the given robot planner. Similarly, McAllister et al. [42] train a model with a weighted loss giving a low weight to the predictions that do not affect the planner. Huang et al. [27] train a forecasting model that allows a simple optimization procedure to select the safest among a set of plans generated by a planner. While prior work considered task-awareness or planner-awareness, to the best of our knowledge, we are the first to use risk as a proxy to make forecasts more useful for the downstream task.

**Subjective probability and prospect theory.** Our pessimistic risk-biased prediction can be interpreted as a model of subjective probability (e.g., [43]), which is closely related to risk-awareness [44]. For instance, prospect theory [45] studies how humans make risk-aware decisions and introduces the notion of *probability weighting* [46]. Under this model, the distribution is "warped" so that the probabilities of unlikely events are always over-weighted. Recent robotics literature has leveraged prospect theory to better model risk-awareness in human decision making, for example, in collaborative human-robot manipulation [47] and driver behavior modeling [48].

Prospect theory is a descriptive model of human decision making, which differs from our goal of designing risk-aware robots. Moreover, our model only overestimates the probability of events that incur high-cost for the robot, unlike probability weighting that overestimates any unlikely outcome.

**Risk-sensitive planning and control.** Risk-sensitive planning and control date back to the 1970s, as exemplified by risk-sensitive Linear-Exponential-Quadratic-Gaussian [49, 50] and risk-sensitive

Markov Decision Processes (MDPs) [51]. More recent methods include risk-sensitive nonlinear MPC [6, 52], Q-learning [44, 53], and actor-critic [54, 55] methods, for various types of risk measures. Refer to a recent survey [56] for further details. Unlike those methods in which the policy directly optimizes a risk-measure, we propose to instead bias the prediction so that risk-sensitivity can be achieved by a risk-neutral planner that simply optimizes the expected value of the cost.

## 3  Background

### 3.1  Generative Probabilistic Trajectory Forecasting

Let $x$ and $y$ be the past and the future trajectories of an agent, and $Y|_x$ denote the random variable of the future trajectory conditioned on the observed past trajectory $x$. We would like to fit the distribution of $p(Y|_x)$ given a dataset $\mathcal{D}$ of i.i.d. samples of $(x, y)$ pairs. To fit $p(Y|_x)$, we maximize the likelihood of future trajectories w.r.t. the model parameters $\theta, \phi$: $\text{maximize}_{\theta, \phi} \prod_{(x,y) \in \mathcal{D}} \mathcal{L}(\theta, \phi; y|_x)$, where $\mathcal{L}(\theta, \phi; y|_x)$ is the likelihood of the sample $y$ knowing $x$. One method to fit this distribution is to learn a conditional variational auto-encoder, CVAE [57]. We focus on this approach because it produces a structured latent representation. The CVAE conditions its likelihood estimation on a latent random variable $Z|_{x,y}$ with a posterior $q_{\phi_2}(z|_{x,y})$, or $Z|_x$ with an inferred prior $q_{\phi_1}(z|_x)$ used in the joint likelihood $p_\theta(y|_x, z|_x)$. The marginal likelihood of the future trajectory (or "model evidence") is $p_\theta(y|_{x,z})$, and can be rewritten as:

$$\mathcal{L}(\theta, \phi; y|_x) \;=\; \int p_\theta(y|_{x,z}) dz \;=\; \int p_\theta(y|_{x,z}) \frac{q_{\phi_2}(z|_{x,y})}{q_{\phi_2}(z|_{x,y})} dz \;=\; \mathbb{E}_{q_{\phi_2}(z|_{x,y})} \left[ \frac{p_\theta(y|_x, z|_x)}{q_{\phi_2}(z|_{x,y})} \right]. \quad (1)$$

Using Jensen's inequality, the logarithm of (1) is lower bounded by

$$L(\theta, \phi; x, y) \;=\; \mathbb{E}_{q_{\phi_2}(z|_{x,y})} \left[ \ln(p_\theta(y|_{x,z})) \right] \;-\; \text{KL}\big(q_{\phi_2}(z|_{x,y}) || q_{\phi_1}(z|_x)\big), \quad (2)$$

called the evidence lower bound (ELBO). We model $q_\phi$ and $p_\theta$ using neural networks. The encoders assume a Gaussian prior with independent elements to produce the inferred prior $f_{\phi_1} : (x) \to (\mu|_x, \text{diag}(\Sigma|_x))$, and the posterior $f_{\phi_2} : (x, y) \to (\mu_{x,y}, \text{diag}(\Sigma|_{x,y}))$. The decoder makes the forecast $g_\theta : (x, z) \to y$. Every term in (2) can be either computed or estimated with Monte-Carlo sampling as established in [57, 58].

### 3.2  Risk Measures

A risk measure is defined as a functional that maps a cost distribution to a real number. In other words, given a random cost variable $C$ with distribution $p$, a risk measure of $p$ yields a deterministic number $r$ called the risk. In practice, we often consider a class of risk measures that lie between the expected value $\mathbb{E}_p[C]$ and the highest value $\sup(C)$. The former corresponds to the risk-neutral evaluation of $C$, while the latter gives the worst-case assessment. Such risk measures often take a user-specified risk-sensitivity level $\sigma \in \mathbb{R}$ as an additional argument, which determines where the risk value $r$ is positioned between $\mathbb{E}_p[C]$ and $\sup(C)$. Formally, let us define a risk measure as $\mathcal{R}_p : (C, \sigma) \to r \in [\mathbb{E}_p[C], \sup(C)]$. Examples of such risk measures include entropic risk [50]: $\mathcal{R}_p^{\text{entropic}}(C, \sigma) = \frac{1}{\sigma} \log \mathbb{E}_p[\exp(\sigma C)]$ as well as CVaR [59]:

$$\mathcal{R}_p^{\text{CVaR}}(C, \sigma) \;=\; \inf_{t \in \mathbb{R}} \left\{ t + \frac{1}{1 - \sigma} \mathbb{E}_p \left[ \max(0, C - t) \right] \right\}. \quad (3)$$

The rest of the paper assumes CVaR (3) as the underlying risk measure, but note that the proposed approach is not necessarily bound to this particular choice. For CVaR, the risk value $r$ given risk-sensitivity level $\sigma \in (0, 1)$ can be interpreted as the expected value of the right $(1 - \sigma)$-tail of the cost distribution [60]. Thus, $\mathcal{R}_p(C, \sigma)$ tends to $\mathbb{E}_p[C]$ as $\sigma \to 0$ and to $\sup(C)$ as $\sigma \to 1$.

Another intriguing property of CVaR is its fundamental relation to distributional robustness. CVaR belongs to a class of risk measures called *coherent measures of risk* [61] with the following dual characterization ([61], Theorem 4a):

$$\mathcal{R}_p(C, \sigma) \;=\; \sup_{q \in \mathcal{Q}} \mathbb{E}_q[C], \quad (4)$$

where $\mathcal{Q}$ is a uniquely-determined, non-empty and closed convex subset of the set of all density functions. This suggests that CVaR is equivalent to a worst-case expectation of the cost $C$ when

the underlying probability distribution $q$ is chosen adversarially from $\mathcal{Q}$. Therefore, an autonomous robot optimizing CVaR (or coherent measures of risk in general) obtains distributional robustness, in that the objective accounts for robustness to potential inaccuracies in the underlying probabilistic model. In this context, the set $\mathcal{Q}$ is often referred to as an *ambiguity set* in the literature [62, 63].

## 4 Problem Formulation

Suppose that a robot incurs cost $C$ under a planned policy $\pi$ or trajectory. This cost is given by a function $J^\pi$ such that $C = J^\pi(Y)$ with $Y$ being the human future trajectory random variable, which the robot predicts probabilistically. We assume that $J^\pi$ is known and differentiable in $y$ for each $\pi$. One can design such a cost function so that $J^\pi(y)$ is high when the robot collides into the particular trajectory $Y = y$ of a human. Supplementary material E defines the cost function used in this work.

We begin with a pre-trained generative model, as defined in Section 3.1, that gives a predictive distribution $p(Y|_x) = \int p(Y|_{x,z})p(z)dz$ through an inferred latent distribution $p(Z|_x)$. This latent is mapped to the trajectory space by a generator or decoder $y = g(z, x)$. Under this unbiased model, the risk is given by $r = \mathcal{R}_p(J^\pi(g(Z, x)), \sigma)$ using the risk measure introduced in Section 3.2.

Given the unbiased model and the risk measure, we are interested in finding another distribution $q_\psi(Z)$ in the latent space with learnable parameters $\psi$, under which simply taking the risk-neutral expectation of the cost will yield the same risk value as given above. This can be achieved by enforcing the following equality constraint on this *biased* distribution $q_\psi(Z)$:

$$\mathbb{E}_{q_\psi}\left[J^\pi(g(Z, x))\right] = \mathcal{R}_p\left(J^\pi(g(Z, x)), \sigma\right). \tag{5}$$

We show that such a distribution exists in Section A.1 of the supplementary material. Comparing both sides in (5), we note that such $q$ should be dependent on the risk-sensitivity level $\sigma$. We propose to optimize the parameters $\psi$ of the risk-biased distribution $q_\psi(Z|_{x,\sigma})$. In general, many distributions $q$ can satisfy (5). We propose to pick a particular $q$ that additionally minimizes the KL divergence from the prior $p$, to prevent the biased distribution from becoming too different from the original unbiased distribution. This leads to the following constrained optimization problem:

$$\underset{\psi}{\text{minimize}} \quad \text{KL}\left(q_\psi(Z|_\sigma)\|p(Z)\right) \quad \text{subject to} \quad \mathbb{E}_{q_\psi}\left[J^\pi(g(Z, x))\right] = \mathcal{R}_p\left(J^\pi(g(Z, x)), \sigma\right). \tag{6}$$

In general, we cannot guarantee uniqueness of the solution to the optimization problem (6). However, in the supplementary material A, we provide further analysis of (6) along with a sufficient assumption under which the solution would be unique (Proposition A.3).

**Connection to importance sampling.** Importance sampling has been employed in rare-event simulation for accelerated safety verification of autonomous systems [64–66], which yields a pessimistic sampling distribution similar to our risk-biased model. However, a crucial difference of our approach is that it estimates a more general risk measure instead of an expected value. Given a desired risk-sensitivity level, unweighted samples from the proposal $q$ will directly yield the risk estimate (5). This removes the need to compute the importance weights.

**Connection to distributional robustness.** When a coherent measure of risk is chosen as the underlying risk measure (such as CVaR), the right-hand side of (5) is always equivalent to a worst-case distribution $q$ chosen out of an ambiguity set $\mathcal{Q}$ (4). In general, it is difficult to verify if the optimal distribution $q_{\psi^*}$ is in $\mathcal{Q}$, since the specifics of $\mathcal{Q}$ depend on the choice of the risk measure as well as the risk-sensitivity level $\sigma$. Nevertheless, it holds true that any feasible distribution $q_\psi$ for (6) yields the same worst-case expected cost as the most adversarial distribution from $\mathcal{Q}$. Therefore, a planner relying on $q_\psi$ instead of $p$ will possess distributional robustness. We demonstrate this crucial capability via an empirical evaluation in Section 6.3.

## 5 Implementation Details

Section B of the supplemental defines a usual (unbiased) CVAE trajectory forecasting model that learns two encoders, defining the Gaussian latent variables $Z|_x$ and $Z|_{x,y}$, and one decoder, predicting $Y|_{x,z}$. We propose to solve problem (6) by learning a third neural network encoder to define

---

**Algorithm 1** Proposed Risk-Biasing Loss Estimation

---

**Input:** Trajectory $(x, y) \sim \mathcal{D}$, risk level $\sigma \sim p(\sigma)$, KL-loss weight $\beta$, risk weight $\alpha$, robot motion $y_{\text{robot}}$
1: **for** $k \in \{1, \ldots, K_1\}$ **do**
2:      Sample latent $z_k|_x \sim \mathcal{N}(\mu|_x, \Sigma|_x)$ with prior parameters $(\mu|_x, \Sigma|_x) = f_{\phi_1}(x)$
3:      Decode risk-neutral predictions $y_k = g_\theta(x, z_k|_x)$
4: Compute risk $r$ using $\{y_1, \ldots y_{K_1}\}$ and $J^{y_{\text{robot}}}$ with Monte Carlo estimation (e.g., [68])
5: **for** $k \in \{1, \ldots, K_2\}$ **do**
6:      Sample biased latent $\hat{z}_k^{(b)} \sim \mathcal{N}(\mu^{(b)}, \Sigma^{(b)})$ with risk-biased parameters $(\mu^{(b)}, \Sigma^{(b)}) = f_\psi(x, \sigma, y_{\text{robot}})$
7:      Decode risk-biased predictions $\hat{y}_k = g_\theta(x, \hat{z}_k^{(b)})$
8: Compute expected cost $\hat{r} = \frac{1}{K_2} \sum_{k=1}^{K_2} J^{y_{\text{robot}}}(\hat{y}_k)$
9: Compute risk loss $L_{\text{risk}} = \rho(\hat{r} - r)$ and prior loss $L_{\text{prior}} = \text{KL}\left(\mathcal{N}(\mu^{(b)}, \Sigma^{(b)}) || \mathcal{N}(\mu|_x, \Sigma|_x)\right)$
**Output:** Loss value $\alpha L_{\text{risk}} + \beta L_{\text{prior}}$ to train $\psi$ ($\theta$ and $\phi_1$ are fixed)

---

a biased latent distribution that, in combination with the pre-trained decoder, produces biased forecasts. This biased encoder takes the past trajectory $x$, a risk-level $\sigma$, and the robot future trajectory $y_{\text{robot}}$. It outputs the parameters of a Normal distribution $\mu^{(b)}$ and $\log(\text{diag}(\Sigma^{(b)}))$.

In practice, we soften the hard constraint (5) by using the penalty method [67], which progressively increases the weight $\alpha$ of the risk-loss during training. We also leverage a user-defined sampling distribution $p(\sigma)$ to sample different risk-sensitivity levels during training, so that the risk estimate remains accurate at any reasonable value of $\sigma$ at inference time. Finally, we encourage the model to overestimate the risk rather than underestimate it so we scale by the positive value $s$ and define an asymmetric risk-loss that penalizes linearly the underestimation of the risk and logarithmically its overestimation:

$$\rho(x) = \begin{cases} s|x|, & \text{if } sx \leq 1 \\ \log(sx), & \text{otherwise.} \end{cases} \tag{7}$$

We obtain the following loss function with $\alpha$ and $\beta$ controlling the relative importance of the losses:

$$\mathcal{L}(\psi) = \mathbb{E}_{\sigma \sim p(\sigma)} \left[ \alpha \rho \left( \mathbb{E}_{q_\psi} \left[ J^\pi(g(Z, x)) \right] - \mathcal{R}_p \left( J^\pi(g(Z, x)), \sigma \right) \right) + \beta \, \text{KL} \left( q_\psi(Z|_{\sigma,x}) \| p(Z|_x) \right) \right].$$

The expected values and the risk measure are approximated by Monte Carlo sampling. For computing CVaR ($\mathcal{R}_p \left( J^\pi(g(Z, x)), \sigma \right)$ ), we use the estimator proposed by Hong et al. [68]. Consistency and asymptotic normality of this estimator hold under mild assumptions [68].

Algorithm 1 lays out the procedure for training our proposed risk-aware prediction. It relies on a fully trained CVAE with the encoder $f_{\phi_1} : x \rightarrow (\mu|_x, \Sigma|_x)$ and decoder $g_\theta : x, z \rightarrow y$ that fits the distribution of $Y|_x$ from a dataset. We train a new latent-biasing encoder $f_\psi : x, \sigma, y_{\text{robot}} \rightarrow (\mu^{(b)}, \Sigma^{(b)})$ to bias the latent distribution while keeping the rest of the CVAE fixed. The risk-level $\sigma$ is randomly sampled on $[0, 1]$ during training and chosen by the user at test time.

## 6 Experiments

### 6.1 Biasing forecasts in a didactic scenario

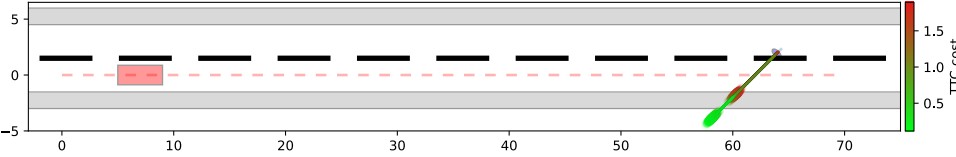

Figure 1: Top-down view of a simulated scene. The robot in red moves left to right down the road as a pedestrian in blue is crossing. The color of the depicted pedestrian trajectory samples indicates their corresponding Time-To-Collision (TTC) cost for the robot. The slow mode in red is more costly than the fast mode in green.

We created the didactic simulation environment in Fig. 1 where a red robot drives at constant speed along a straight road with a stochastic pedestrian. The pedestrian either walks slowly or quickly, yielding a bimodal distribution over their travel distance. We collected a dataset in this environment where the initial position and orientation of the pedestrian are set at random. We used it to train a risk-biased CVAE model according to the method presented in sections 4 and 5. Fig. 2b shows

the risk-neutral prediction ($\sigma = 0$) of the pedestrian's travel distance in a specific scene. As can be seen, the model captures both of the equally-likely modes. In contrast, in the risk-biased case ($\sigma = 0.95$), the model predicts the slower mode with much greater frequency because, in this scene, if the pedestrian walks slowly it will collide with the robot. If, alternatively, the pedestrian walks quickly, the vehicle will pass behind it safely without collision. In other words, the risk-biased model pessimistically predicts collisions with a greater probability than does the risk-neutral model. With $\sigma = 0.95$, pessimism is so high that the safer mode falls in the tail of the distribution. In supplemental D we explore the latent representation of this model.

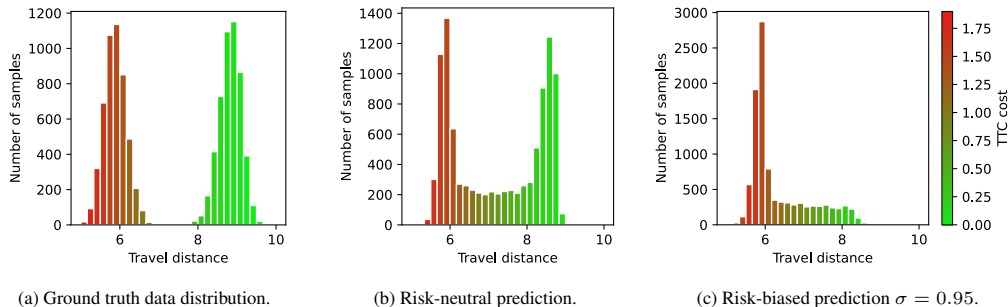

(a) Ground truth data distribution.  (b) Risk-neutral prediction.  (c) Risk-biased prediction $\sigma = 0.95$.

Figure 2: Histograms of the pedestrian travel distances at the end of the 5-second episode in the defined scene. Each bar is colored with the average Time-To-Collision (TTC) cost of the bin.

## 6.2 Planning with a biased prediction

The previous experiment demonstrates the ability of our approach to bias predictions towards dangerous outcomes. The experiment in this section evaluates whether this ability can benefit online planning using a model-based trajectory optimization algorithm. In this setting, predictions are used to evaluate the risk of various candidate robot trajectories, $y_{\text{robot}}$, in order to select the best one.

We generated a new dataset in which the robot's initial speed and per-timestep accelerations are sampled randomly, as opposed to the constant velocity model used previously. This variation ensures that the robot trajectories generated by the planner are within the training distribution. We modified the biasing encoder to account for the changing robot trajectories by adding $y_{\text{robot}}$ to its inputs. This allows our new model to achieve pessimistic forecasting with respect to a particular $y_{\text{robot}}$.

The online planner controls the longitudinal acceleration of the robot, which is modeled as a double integrator system. We employed the cross entropy method (CEM) [69, 70] as the underlying optimization algorithm. CEM is a stochastic optimization method that locally optimizes $y_{\text{robot}}$ based on a given initial $y_{\text{robot}}^{\text{init}}$. In each episode, CEM first draws $n_{\text{samples}}$ risk-biased prediction samples given $y_{\text{robot}}^{\text{init}}$ and observed pedestrian motion $x$, and uses the samples to produce a locally optimal $y_{\text{robot}}^*$. This process involves a linear complexity in $n_{\text{samples}}$. We incorporated a quadratic trajectory tracking cost in the planner's objective so that the robot is encouraged to continue moving at a constant speed when the risk of collision is deemed negligible. See supplementary material C.1.2 for details.

We evaluated the performance of the combined risk-biased predictor and CEM planner across 500 episodes. Fig. 3 shows that CEM using the risk-biased predictor consistently produces $y_{\text{robot}}^*$ with low Time-To-Collision (TTC) cost values, even when sampling few trajectories. We compared with a baseline approach in which a risk-sensitive version of CEM performs planning using an unbiased CVAE predictor. We obtained this risk-sensitive planner by replacing the Monte Carlo expectation with the CVaR estimator [68]. This baseline is an instance of the conventional risk-sensitive planning with data-driven human motion forecasting [6, 7], which evaluates the risk within the planner rather than in the predictor. Fig. 3 shows significantly higher TTC cost of $y_{\text{robot}}^*$ for the baseline when CEM uses fewer than 16 samples. This is because the collision risk is underestimated with few samples, and thus the planner over-optimistically optimizes trajectory tracking to the detriment of safety.

## 6.3 Robustness to out-of-distribution pedestrian behavior

For this last didactic experiment, we evaluated the distributional robustness of the overall prediction-planning pipeline to a test-time change of the pedestrian stochastic behavior model. Specifically, we used the same dataset and planner as in Section 6.2, but we reduced the overall average speed of the pedestrian by 25% *only at test time, after training*. Other factors such as bi-modality were

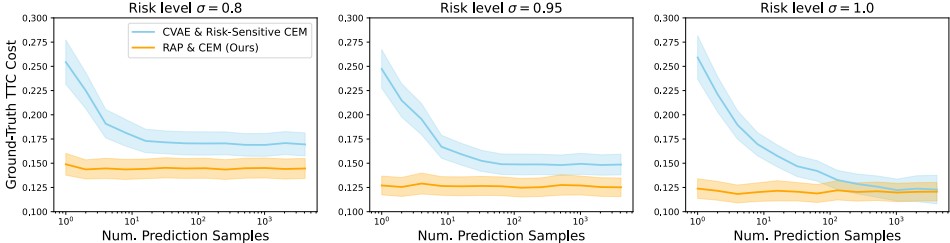

Figure 3: Ground-truth TTC cost of the optimized $y_{robot}^*$, averaged over 500 episodes (lower the better). Ribbons show 95% confidence intervals of the mean. Our risk-biased predictor (RAP) coupled with CEM consistently achieves low cost regardless of the number of prediction samples that CEM draws online from the predictor.

Table 1: Ground-truth TTC cost of the optimized $y_{robot}^*$ under different test-time pedestrian behaviors, averaged over 500 episodes (lower the better).

| Predictive Model | # Prediction Samples | Planner | $\sigma$ | Pedestrian Behavior | TTC Cost |
|---|---|---|---|---|---|
| Unbiased CVAE | 64 | Risk-Neutral CEM | NA | same as training | $0.23 \pm 0.01$ |
| Unbiased CVAE | 64 | Risk-Neutral CEM | NA | 25% reduced speed | $0.44 \pm 0.02$ |
| Unbiased CVAE | 64 | Risk-Sensitive CEM | 0.95 | 25% reduced speed | $0.37 \pm 0.02$ |
| Biased CVAE (RAP) | 64 | Risk-Neutral CEM | 0.95 | 25% reduced speed | $0.34 \pm 0.02$ |
| Unbiased CVAE | 1 | Risk-Sensitive CEM | 0.95 | 25% reduced speed | $0.46 \pm 0.02$ |
| Biased CVAE (RAP) | 1 | Risk-Neutral CEM | 0.95 | 25% reduced speed | $0.34 \pm 0.02$ |

kept the same as at training time. From the robot's perspective, reducing the average speed of the pedestrian in the test scenario (as exemplified in Fig. 1) results in an adversarial distribution shift. We studied how robust the risk-aware robot is under this out-of-distribution pedestrian behavior that the predictor did not witness during training.

Table 1 summarizes the results of this experiment. The first row shows the nominal in-distribution risk-neutral case. The second row shows that the risk-neutral robot is not robust at all to the distribution shift, resulting in the doubling of the nominal TTC cost. This is expected from an autonomy stack without risk-awareness. Rows 3 and 4 suggest that risk-awareness improves robustness given sufficient samples from the predictor. Finally, the last two rows show that our proposed framework remains robust to the distribution shift *even with a single prediction sample*, whereas the conventional risk-sensitive prediction-planning approach (row 5) does not show robustness any more. This demonstrates that, when the computation budget for online planning is limited, distributional robustness is better achieved within the predictor rather than in the planner.

## 6.4   Application to real-world data

We applied our risk-biased forecasting framework to real-world data from the Waymo Open Motion Dataset (WOMD) [71]. Following a similar approach to [27], we selected the annotated scenarios that cover interesting interactions between two agents. We randomly selected one of the two interacting agent as the ego and the other as the agent to predict. We input other agent tracks and the map as additional conditioning information to account for the interaction with the environment and the other agents. Then, we trained a biased CVAE model as described in Section 5. In this experiment we only conditioned the biased-encoder on the ego past trajectory, not its future trajectory, in order to avoid ground truth information leakage. This means that the biased-encoder is making an implicit forecast of the ego future, which leads to the failure mode presented in Fig. 4d wherein the wrong implicit forecast leads to an incorrectly-biased distribution.

Table 2 shows the results of this experiment. First note the large difference between the minFDE and FDE values of the unbiased model, which illustrates that the predictions are diverse. This is qualitatively supported by Fig. 4, which shows a wide diversity of predicted trajectories. As expected, our biased CVAE model (RAP) with a risk level $\sigma = 0$ shows results that match the ones from the unbiased model. As the risk-level increases, the predictions increasingly differ from those of the unbiased distribution as well as the ground truth trajectory. This is reflected by the larger minFDE and FDE values. At $\sigma = 1$, the biased prediction distribution collapses to the mode that the model estimates to be the most costly, yielding minFDE close to FDE. The risk estimation error is the average difference of the mean *cost* under the biased prediction and the *risk* estimated using a large number of samples from the unbiased prediction. Its mean value shows the risk estimation bias of the proposed approach while its mean *absolute* value shown in the next column shows the average error that is made in either direction. Up to $\sigma = 0.95$, the risk estimation is nearly unbiased.

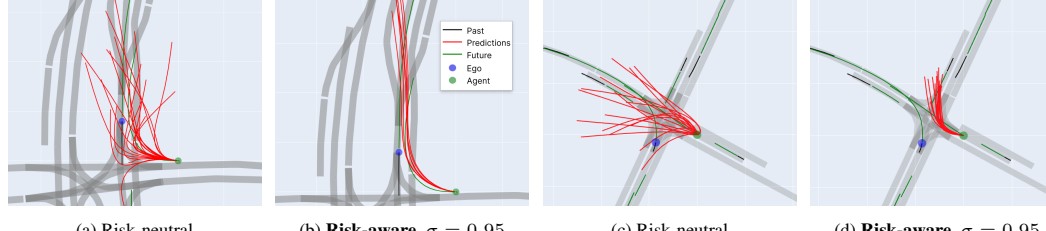

| (a) Risk-neutral | (b) **Risk-aware**, $\sigma = 0.95$ | (c) Risk-neutral | (d) **Risk-aware**, $\sigma = 0.95$ |

Figure 4: Visualization of WOMD scenes and forecasts. The lane centerlines are represented in gray, the past observations in black, the ground truth futures in green, and the 16 forecast samples in red. The ego is in blue and the agent to predict in green. In figure **(b)** the risk-aware forecasts turns towards the ego. In figure **(d)** the risk-aware forecasts would be costly if the ego went straight ahead. This is a failure case where the forecasts are biased towards an expected ego trajectory that did not occur.

Table 2: Motion forecasting error and risk estimation error on the WOMD validation set. **minFDE (K)**: minimum final displacement error over K samples, **risk error (K)**: mean value of the signed difference between the average cost of the biased forecasts over K samples and the risk estimation using the unbiased forecasts, **risk |error| (K)**: mean value of the absolute values of the risk estimation error.

| Predictive Model | $\sigma$ | minFDE(16) | FDE (1) | Risk error (4) | Risk \|error\| (4) |
|---|---|---|---|---|---|
| Unbiased CVAE | NA | $3.82 \pm 0.04$ | $13.06 \pm 0.01$ | NA | NA |
| Biased CVAE (RAP) | 0 | $3.81 \pm 0.04$ | $13.07 \pm 0.02$ | $0.00 \pm 0.00$ | $0.12 \pm 0.00$ |
| Biased CVAE (RAP) | 0.3 | $4.32 \pm 0.05$ | $11.89 \pm 0.02$ | $0.02 \pm 0.00$ | $0.13 \pm 0.00$ |
| Biased CVAE (RAP) | 0.5 | $5.32 \pm 0.06$ | $12.05 \pm 0.02$ | $0.02 \pm 0.00$ | $0.16 \pm 0.00$ |
| Biased CVAE (RAP) | 0.8 | $7.78 \pm 0.07$ | $13.53 \pm 0.02$ | $0.01 \pm 0.00$ | $0.26 \pm 0.00$ |
| Biased CVAE (RAP) | 0.95 | $10.13 \pm 0.09$ | $15.33 \pm 0.02$ | $0.03 \pm 0.01$ | $0.43 \pm 0.00$ |
| Biased CVAE (RAP) | 1 | $11.58 \pm 0.09$ | $16.29 \pm 0.02$ | $-0.22 \pm 0.01$ | $0.60 \pm 0.01$ |

At $\sigma = 1$, the risk estimation is slightly under-estimated. Usage of real data in this section limits our ability to provide accurate safety statistics for our approach. Instead, we provide extensive qualitative results. Section C of the supplement gives additional experimental details and results. We also provide extra experiments, figures and animations on our project website[4]. Finally, the model can be tested directly on several hundred samples, with any risk-level, on our online demo[5].

## 7  Limitations

A first limitation of our approach is that the constrained optimization problem (6) is difficult to solve in practice due to challenges presented in Section 4. The constraint relaxation and the neural network optimization method yield a sub-optimal risk-aware predictor that may still underestimate risk when the risk-sensitivity level is close to 1. Therefore, our method might be inadequate when an extremely conservative behavior is desired, or in the case of extremely low probability but catastrophic events. Second, *in the real data application*, our risk-aware prediction is not conditioned on a specific robot plan which may lead the forecast to collapse onto a mode that is not the most critical. Finally, we only forecast marginal agent behavior instead of jointly predicting the behaviors of several agents in the scene. This neglects potential risk-avoiding interactions in the future and leads to overly pessimistic biased forecasts.

## 8  Conclusion

This paper proposes a risk-aware trajectory forecasting method for robust planning in human-robot interaction problems. We present a novel framework to learn a pessimistic distribution offline that simplifies online risk evaluation to expected cost estimation. Our experimental results show that this method leads to safe robot plans with reduced sample complexity. We additionally demonstrate the effectiveness of our approach in real-world scenarios with low risk-estimation error and strong qualitative results. In future work, we intend to evaluate our approach in a realistic simulator, and also improve the accuracy of risk estimation on real-world data by conditioning biased prediction on potential robot plans.

---

[4] https://sites.google.com/view/corl-risk/home
[5] https://huggingface.co/spaces/TRI-ML/risk_biased_prediction

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
