# OpenReview forum: "RAP: Risk-Aware Prediction for Robust Planning"
_robot-learning.org/CoRL/2022/Conference — CoRL 2022 Oral_

### Official Review · Reviewer_zkWD · 2022-07-14

**Originality:** Good
**Technical Quality:** Very Good
**Clarity Of Presentation:** Very Good
**Impact:** 4

**Recommendation:**

Weak Accept: I recommend accepting the paper, but will not argue for my recommendation if the majority of other reviewers have a different opinion.

**Summary:**

This paper tackles risk sensitive planning for vehicle prediction. Existing approaches often fail make diverse predictions suitable for risk aware planning. Additionally, adapting an existing risk-neutral planning pipeline to risk aware one may be costly.

Instead of targeting downstream risk-sensitive control, this paper encodes risk into upstream prediction components. The authors' key insight is that a risk sensitive measure of cost is the same as the risk-neutral expectation of cost under a distribution of trajectories with risky trajectories up-weighted. This means they can use VAEs to learn an importance sampler which biases towards risky parts of the trajectory space, implicitly learning a risk-sensitive measure via proxy.

Experiments are given on a small pedestrian proof of concept scenario, as well as larger experiment on the waymo open dataset. They demonstrate usefulness of their method as compared to an unbiased CVAE.

**Issues:**

No major issues. As mentioned in the strengths/weakness section, any insights you could provide into how this work fits into the context of existing rare-event sampling literature, and intuitions about how some of the hyper-parameters in the experiments section affect performance, would help broaden my understanding of the work.

**Quality Of The Limitations Section:**

Limitations are addressed clearly

**Reviewer Expertise:**

3: The reviewer is fairly confident that the evaluation is correct

**Robotics Focus:**

Highly relevant to robotics but no hardware experiments

**Strengths And Weaknesses:**

## Strengths

- The paper is easy to follow, with good mathematical layout and comprehensive existence proofs accompanying each of the measure formulae.
- The contribution of the paper itself is a novel synthesis of Variational Autoencoder methods, Cross-Entropy importance sampling, and risk sensitive measures. This combination provides a valuable tool for adding robustness to the predictive layers of an perceptual-control pipelines.
- The results and figures given (in particular, the histograms of sampling for the biased vs unbiased experiments) do a good job showing the intuitive operation of their algorithm, and how it effectively biases predictive distributions towards risky parts of the space.

## Weaknesses

-  There is an odd statement towards the end of section 4 about proving that the solution to their optimization problem is unique. This is under the assumption that the decoder is "Volume Preserving" and they even provide a citation for a method for implementing volume-preserving neural networks. However, they then state that they do not attempt to meet this assumption, yet still present it as evidence their problem is well posed. I'm not sure I follow this! I think for the statement to be of any value, I would as least like to intuitively understand what it would take to bring their method in line with the volume preservation assumption.
- In Section 5.2, there are a few user-defined constants ('s', 'alpha', 'beta'). It would be nice to see some analysis of how altering these values affects the end result.
- The work is well situated within most of the related literature it deals with. However, the methods used also draw heavily on importance sampling and rare-event validation. I would have liked to see more engagement with this work. For instance, how does the work fit alongside things like (https://ieeexplore.ieee.org/abstract/document/7933977) or (https://proceedings.neurips.cc/paper/2018/hash/653c579e3f9ba5c03f2f2f8cf4512b39-Abstract.html)? Additionally, how do their methods relate to concepts like the proven ideal optimal zero-variance sampler from the importance sampling literature?
- The authors mention in their limitations section that, while their method works for risk-aware planning at the 95% level, it may not be suitable for much rarer events. I agree, but I think this point is understated: For the AV domain in which their experiments take place, *extremely rare collision events are, in fact, the norm*, with safety requirements several magnitudes stronger than the ones given in the paper.

**Summary Of Recommendation:**

On the basis of the strengths and weaknesses outlined above, I think this is a good candidate for publication. My suggestions for additional analysis and literature are minor issues, stemming from a desire to see the work expanded, rather than anything preventing the work from achieving publication quality.

---

> ### Author Response · Authors · 2022-08-23
> **Response to Reviewer zkWD (1/2)**
>
> **Comment:**
>
> We would like to thank you for taking the time to review our paper. We greatly appreciate your constructive feedback, especially the one commenting on importance sampling for rare-event simulation. We will address the questions and concerns raised below. **We have also made further points in the general response to all the reviewers in a separate thread. Please see "General Response 2/2" for the updated paper and the supplementary materials.**
>
> > There is an odd statement towards the end of section 4 about proving that the solution to their optimization problem is unique. This is under the assumption that the decoder is "Volume Preserving" and they even provide a citation for a method for implementing volume-preserving neural networks. However, they then state that they do not attempt to meet this assumption, yet still present it as evidence their problem is well posed. I'm not sure I follow this! I think for the statement to be of any value, I would as least like to intuitively understand what it would take to bring their method in line with the volume preservation assumption.
>
> This statement was somewhat stretching the conclusions that could be drawn from our analysis of the uniqueness problem, so we rephrased it. We also removed the unnecessary mention of volume preservation from the main text.
>
> > In Section 5.2, there are a few user-defined constants ('s', 'alpha', 'beta'). It would be nice to see some analysis of how altering these values affects the end result.
>
> $s$ is chosen such that the risk estimation error is close to 0 on average while not underestimating the risk on too many samples. We provide a graph of the risk estimation error during training for different values of $s$ in our didactic experiment. (Please download the zip file attached to this thread.) A curve like the one for $s=150$ seems appropriate: the model overestimates the risk at first and progressively learns a better fit that is closer to an unbiased risk estimation.
> We chose a beta value empirically such that the reconstruction is good and the min error over samples of the inference distribution has a similar value which shows a good inference diversity. The effect of beta in CVAEs is discussed in appendix A.3 of https://arxiv.org/pdf/2007.12036.pdf. Alpha can be normalized out of the loss and compensated with a different learning rate.
> Our model relies on many other hyper-parameters but we did not find any that needed to be set very precisely or that had an unexpected effect.
>
> > The work is well situated within most of the related literature it deals with. However, the methods used also draw heavily on importance sampling and rare-event validation. I would have liked to see more engagement with this work. For instance, how does the work fit alongside things like (https://ieeexplore.ieee.org/abstract/document/7933977) or (https://proceedings.neurips.cc/paper/2018/hash/653c579e3f9ba5c03f2f2f8cf4512b39-Abstract.html)? Additionally, how do their methods relate to concepts like the proven ideal optimal zero-variance sampler from the importance sampling literature?
>
> Thank you for bringing those papers to our attention. We gave an extensive review to those lines of work, and added a paragraph in Section 4 where we cite the suggested papers and clarify differences of our work from importance sampling. Indeed, our work shares similarities to importance sampling for rare-event simulation, in that it produces a biased sampling distribution that focuses more on dangerous events. However, a crucial theoretical difference is that the quantity to be estimated is a risk measure, which is more general than the expected value. (Note that the rare event probability is equal to the expected value of the indicator function corresponding to the rare event.) This generalization in our approach removes the need for computing the importance weights; given a desired risk-level, unweighted samples from the proposal distribution will directly give the risk estimation.
>
> With regard to the zero-variance sampler, Figure 13 in supplementary material D indicates that the variance tends to decrease as the risk-level increases. This observation implies that higher risk levels will bring the distribution closer to a zero-variance sampler, since the prediction is focused on the most dangerous outcome. On the other hand, when the risk-level is small or zero, we would prefer a sampling distribution with a large variance that would offer a better coverage of the possible agent behaviors. Intermediate risk levels are expected to smoothly move from one regime to the other.
>
> (Continued to Response to Reviewer zkWD 2/2)
>
> **Zip File:**
>
> /attachment/7059a73c63b96c858a0fd7c38bf67cf3076762b5.zip

---

> > ### Author Response · Authors · 2022-08-23
> > **Response to Reviewer zkWD (2/2)**
> >
> > (Continued from Response to Reviewer zkWD 1/2)
> >
> > > The authors mention in their limitations section that, while their method works for risk-aware planning at the 95% level, it may not be suitable for much rarer events. I agree, but I think this point is understated: For the AV domain in which their experiments take place, extremely rare collision events are, in fact, the norm, with safety requirements several magnitudes stronger than the ones given in the paper.
> >
> > With the CVaR risk, at a 95% risk-level, the 5% tail of the most costly events is used. The extremely rare events should be represented in that distribution tail but might still be under-represented. The quality of the underlying unbiased forecasting model might not allow the capture of extremely rare events. Making a better unbiased predictive model would be needed with either approach to risk estimation. Future work could attempt to train a biased predictive model without the need for an underlying unbiased predictor. This might allow a better capture of the costly rare events.
> >
> > We thank you again for your careful review. Please let us know if this addresses all of your questions and concerns.

---

### Official Review · Reviewer_vmFo · 2022-07-25

**Originality:** Very Good
**Technical Quality:** Very Good
**Clarity Of Presentation:** Very Good
**Impact:** 3

**Recommendation:**

Weak Accept: I recommend accepting the paper, but will not argue for my recommendation if the majority of other reviewers have a different opinion.

**Summary:**

There are several recent works that aim to enhance robustness of planning against various sources of uncertainty. This is typically accomplished by  designing risk-aware planning methods. The issue in these methods is that is risk is approximately computed which may lead to unsafe behaviors. This paper follows a different approach and instead of robustifying planning, it robustifies prediction by making it risk-aware. Particularly, a new prediction objective is proposed to learn risk-biased distribution over trajectories of agents that exist in the environment. This approach reduces sample-complexity during online planning which facilitates real-time implementation. The proposed method is supported by simulation results.



**Issues:**

The authors should address the following issues:

- The evaluation needs to be improved to demonstrate distributional robustness in a more convincing way. For instance, what happens if the human behaves differently between training and test time (section 6.2)? In this scenario, how does the proposed method perform and also how is its performance compared against risk-sensitive planners?

- Figures demonstrating the robot trajectories generated by the proposed method as well as competitive approaches would be helpful (Section 6.2). In fact, this would help visualize how conservative the designed paths are.

- In the simulation studies, the TTC metric needs to be defined. Also, how are the prediction samples defined (e.g., in Fig. 3)? Do they refer to sample trajectories using to train the predictor?

- In the abstract, the authors state :"This reduces sample complexity during online planning for safe real-time performance". This statement is confusing. How is sample complexity related to online planning? Samples are collected offline to train a risk-biased predictor. Given, this trained predictor, a planner is used online to design trajectories. Even if the offline training process is not sample efficiency, this has nothing to do with online planning, since samples are not used online for training. The authors should re-write this sentence more clearly.




**Quality Of The Limitations Section:**

Limitations are addressed clearly

**Reviewer Expertise:**

3: The reviewer is fairly confident that the evaluation is correct

**Robotics Focus:**

Highly relevant to robotics but no hardware experiments

**Strengths And Weaknesses:**

Strengths:

The idea of proposing a risk-biased trajectory forecasting framework is quite interesting and novel. This pessimistic forecasting allows the planner to be distributionally robust, i.e., robust to un-modeled sources of uncertainty.

The proposed method does not require modification of the planner and in fact it seems that it can be integrated with various planners (e.g., risk-aware planners and risk-neural planners).

A biased distribution  is constructed from which samples are drawn to forecast future states of agents. This biased distribution allows to accurately predict future states with only few samples.

Weaknesses

The evaluation is somewhat weak. It should be improved to demonstrate more clearly the distributional robustness of the proposed method.




**Summary Of Recommendation:**

Overall, the authors propose a novel and quite interesting approach to robustify predictions. The proposed risk-aware forecaster can be coupled with existing planners, whether they are risk-aware or risk-neural, and it can design safer (but more conservative) paths than existing risk-sensitive planners. The developed algorithm is supported by nice simulation results. Nevertheless, the evaluation should be improved to demonstrate distributional robustness in a more convincing way.

---

> ### Author Response · Authors · 2022-08-23
> **Response to Reviewer vmFo**
>
> Thank you for taking the time to review our paper. We highly appreciate your constructive comments, especially the one suggesting to test distributional robustness for improved evaluation. We will address the questions and concerns raised below. **We have also made further points in the general response to all the reviewers in a separate thread. Please see "General Response 2/2" for the updated paper and the supplementary materials.**
>
> > The evaluation needs to be improved to demonstrate distributional robustness in a more convincing way. For instance, what happens if the human behaves differently between training and test time (section 6.2)? In this scenario, how does the proposed method perform and also how is its performance compared against risk-sensitive planners?
>
> We fully agree that the evaluation should include a demonstration of distributional robustness, so we decided to conduct the suggested experiment and reported the results in Section 6.3, a newly added section. Table 1 in the revised paper clearly shows the benefit of our approach over both risk-neutral (thus non-robust) framework as well as the risk-sensitive planning baseline. Our proposed approach is the only one that still remains robust to the test-time distribution shift of the pedestrian behavior, when only a single prediction sample is given to the online planner.
> We would like to thank you for your valuable suggestion, as we believe that this additional experiment greatly improves the consistency and the strengths of our work.
>
> > Figures demonstrating the robot trajectories generated by the proposed method as well as competitive approaches would be helpful (Section 6.2). In fact, this would help visualize how conservative the designed paths are.
>
> Due to space limits, we decided to add the requested figures in Section C.1.2 of the supplementary material. Please see Figure 10, in which the risk-sensitive planner baseline almost collided with the pedestrian whereas our approach maintained more distance. We also included an animated version of those road scenes in the zipped supplementary materials and on our website (https://sites.google.com/view/corl-risk/planning).
>
> > In the simulation studies, the TTC metric needs to be defined. Also, how are the prediction samples defined (e.g., in Fig. 3)? Do they refer to sample trajectories using to train the predictor?
>
> The TTC cost is defined in section E of the supplementary material.
>
> For the question regarding the sample trajectories, please also see our response to your other question below. “Prediction Samples” in Figure 3 refers to the samples that the planner draws online from the predictor, for internally evaluating and optimizing the robot’s trajectory. They are not the same as the samples needed for offline training, and we do not claim to improve the sample efficiency of the training process (e.g., Algorithm 1).
>
> > How is sample complexity related to online planning? Samples are collected offline to train a risk-biased predictor. Given, this trained predictor, a planner is used online to design trajectories. Even if the offline training process is not sample efficiency, this has nothing to do with online planning, since samples are not used online for training. The authors should re-write this sentence more clearly.
>
> Thank you for your question. We have revised the corresponding part in the paper. In fact, samples are drawn online (i.e., at deployment time) from the predictor and given to the planner, because the planner has to evaluate the risk objective on-the-fly, using those samples that capture stochasticity of future human motion. This is not unique to our work; prior literature (for instance https://arxiv.org/abs/1710.09483 and https://arxiv.org/abs/2009.05702) also requires online sampling from the predictor for evaluating and optimizing the trajectory of the robot. Those samples are not the same as the samples used for offline training, as the network weights of the predictor are already fixed during online deployment.
>
> We thank you again for your careful review. Please let us know if this addresses all of your questions and suggestions.

---

> > ### Comment · Reviewer_vmFo · 2022-08-25
> > **Response to authors**
> >
> > Thanks for carefully addressing my comments. I recommend the revised paper for publication.

---

> > > ### Author Response · Authors · 2022-08-28
> > > **Response to Reviewer vmFo**
> > >
> > > Thank you for the acknowledgement of our response. If you believe this work would be of interest to the CoRL community, we ask that you increase your score in Part 2 of the review process, so as to increase the visibility of the paper.

---

### Official Review · Reviewer_QxWd · 2022-07-31

**Originality:** Good
**Technical Quality:** Very Good
**Clarity Of Presentation:** Excellent
**Impact:** 4

**Recommendation:**

Weak Accept: I recommend accepting the paper, but will not argue for my recommendation if the majority of other reviewers have a different opinion.

**Summary:**

This paper proposes to learn a risk-aware prediction model. The key idea is to learn an extra encoder model of the latent random distribution minimizing a cost function biasing the latent random variable to generate risk-biased trajectories. The article makes two main contributions:
1 - a risk-aware prediction model
2 - a novel training objective to learn a risk-biased distribution of the latent random variable

**Issues:**

Presented results only consider simple scenarios with only 2 agents.
Therefore, in future work and in a journal of this work I recommend the authors to:
1 - Consider complex scenarios with a high number of agents.
2 - Evaluate and discuss how this approach would scale to higher number of agents
3 - Will over-conservative prediction lead to the freezing-robot problem? How to address this issue?

**Quality Of The Limitations Section:**

Additional details required

**Reviewer Expertise:**

4: The reviewer is confident but not absolutely certain that the evaluation is correct

**Robotics Focus:**

Highly relevant to robotics but no hardware experiments

**Strengths And Weaknesses:**

First, this paper tackles an extremely challenging and important problem to the robotics community. Secondly, the proposed idea is interesting and the proposed approach novel. Overall, the paper is well written, the method is described clearly and the presented results support the claimed contributions.

However, this paper has the following main limitations
1 - It only considers simple scenarios with a maximum of 2 agents
2 - the risk-aware prediction do not depend on the ego's plan

Therefore, in future work and in a journal of this work I recommend the authors to:
1 - Consider complex scenarios with a high number of agents.
2 - Evaluate and discuss how this approach would scale to higher number of agents
3 - Will over-conservative prediction lead to the freezing-robot problem? How to address this issue?

The presented results can also be strengthened:
1 - Present more qualitative results in the form of a video to clearly show the how the predictions vary with the risk value
2 - How computationally demanding is to train such model? What is the training time? How much time it takes on training iteration?
3 - How does this model generalizes to new unseen scenarios?

Finally, I recommend the author to revise their citations. It is good that they did such broad literature review but there are several sentences where the citation does not support the written sentence.

**Summary Of Recommendation:**

This paper tackles and important problem and it proposes an interesting new method. The presented results only consider very simple scenarios but I am ok with those for a conference paper as they clearly demonstrate the advantage of the proposed approach. Adding more experimental results, in a video format for instance, would improve the contribution of the article. Moreover, the limitations section needs to address more issues of the proposed approach. Overall, i think it is a good paper, well written and it has a clear contribution which the results support. Therefore, I recommend its publication.

---

> ### Author Response · Authors · 2022-08-23
> **Response to Reviewer QxWd**
>
> We would like to thank you for taking the time to review our paper. The constructive comments provided have greatly helped us to revise the manuscript. We will address the questions and concerns raised below. **We have also made further points in the general response to all the reviewers in a separate thread. Please see "General Response 2/2" for the updated paper and the supplementary materials.**
>
> > this paper has the following main limitations 1 - It only considers simple scenarios with a maximum of 2 agents 2 - the risk-aware prediction do not depend on the ego's plan
>
> Thank you for your feedback. The issue you raised about the limited number of agents considered was also pointed out by the area chair and we addressed it in our general response:
>
> Sophisticated multi-agent scenarios are considered in our Waymo open motion dataset (WOMD) application. Specifically, while only pairwise risk between the ego and one other agent is accounted for, the context used to make the forecast includes other agents and the local map. This means that our method is able to handle marginal biased predictions for each agent in the scene, but indeed not jointly-biased predictions of the whole scene. Using marginal distributions would probably lead to overly pessimistic predictions, as potential risk-avoiding interactions would be neglected. Biasing the joint prediction would be an extension of our work and could be done with our framework, for instance by using a multi-agent cost function and a joint generative forecasting model.
>
> We added this consideration in the limitations section 7.
>
> > in future work and in a journal of this work I recommend the authors to: 1 - Consider complex scenarios with a high number of agents. 2 - Evaluate and discuss how this approach would scale to higher number of agents 3 - Will over-conservative prediction lead to the freezing-robot problem? How to address this issue?
>
> We single out the vehicle pair that is labeled as having an active interaction in each scene to over-sample higher risk situations but we do condition the predictions on all the other agents. We believe that most of the other vehicle pairs are in a low-risk situation with respect to the ego and would not be as interesting to consider for a risk estimation evaluation.
> Handling joint predictions of several agents in the scene is challenging. However, if captured correctly, this would allow less conservative plans to be considered in some situations.
>
> Overly-pessimistic predictions could lead to the freezing-robot problem. One solution to avoid over-conservative behaviors would be to lower the risk-sensitivity level. As in prior literature in risk-aware planning and control, this work assumes that the risk-sensitivity level is a user-specified input. Alternatively, recent papers, such as https://las.inf.ethz.ch/files/trautman10unfreezing.pdf and  https://arxiv.org/pdf/2104.10558.pdf  show that predictions that consider other agents’ response to the ego’s action allow the system to avoid some overly conservative behavior.
>
> > The presented results can also be strengthened: 1 - Present more qualitative results in the form of a video to clearly show the how the predictions vary with the risk value 2 - How computationally demanding is to train such model? What is the training time? How much time it takes on training iteration? 3 - How does this model generalizes to new unseen scenarios?
>
> We provided more qualitative results in the zipped supplementary materials as well as on a website: https://sites.google.com/view/corl-risk/home. Among these, we made a number of clips showing the forecasts for fixed random samples as the risk-level is increased.
> Complexity analyses are provided in the supplementary material, for both offline model learning and online planning. (Supplementary B, C.1.2)
>
> Regarding generalization of our model, we do not expect the risk-biased predictive model itself to accurately extrapolate its predictions to out-of-distribution (OOD) scenes. However, an additional experiment that we added in Section 6.3 indicates that the overall prediction-planning pipeline remains robust to the test-time distribution shift of the pedestrian behavior. This is due to the distributional robustness property of CVaR, as discussed in detail in Section 4. In this sense, our risk-biased predictive model is still useful for the downstream planning task in OOD scenes, even though the prediction itself may not be accurate.
>
> > It is good that they did such broad literature review but there are several sentences where the citation does not support the written sentence.
>
> We have extensively checked the references and revised ambiguous ones that we were able to identify. If you have specific remaining concerns and suggestions regarding references, we would greatly appreciate further clarification.
>
> We thank you again for your careful review. Please let us know if this addresses all of your suggestions and questions.

---

> > ### Comment · Reviewer_QxWd · 2022-08-25
> > **Response to Authors**
> >
> > Thanks for revising the manuscript. I recommend the current version of the article for publication.

---

> > > ### Author Response · Authors · 2022-08-28
> > > **Response to Reviewer QxWd**
> > >
> > > Thank you for the acknowledgement of our response. If you believe this work would be of interest to the CoRL community, we ask that you increase your score in Part 2 of the review process, so as to increase the visibility of the paper.

---

### Official Review · Reviewer_UeEy · 2022-08-08

**Originality:** Very Good
**Technical Quality:** Excellent
**Clarity Of Presentation:** Excellent
**Impact:** 4

**Recommendation:**

Strong Accept: I recommend accepting the paper and will argue for my recommendation even if other reviewers hold a different opinion.

**Summary:**

The paper considers risk estimation of predictors used by planners for robots. Robustness of plans is achieved by pushing the risk estimation to prediction, rather than computing it at planning time. This has two advantages: (a) planners do not need to be updated when considering new predictors and risk measures; and (b) it simplifies planning using expectations over biased (risk-aware) distributions. The paper shows the approach on didactic and real-world datasets.

**Issues:**

In Sec. 5.2, what is $s$? Should $s = \sigma$ or is $s \geq 1$?
Moreover, the penalty $\rho(x)$ is discontinuous in $x$. How does it impact performance? Also, is the problem well posed?  Would $s \cdot (1 + \log(x) )$ work instead of $\log(sx)$?

**Quality Of The Limitations Section:**

Limitations are addressed clearly

**Reviewer Expertise:**

4: The reviewer is confident but not absolutely certain that the evaluation is correct

**Robotics Focus:**

Highly relevant to robotics but no hardware experiments

**Strengths And Weaknesses:**

Strengths
+ The paper is clear and well written.
+ It has good explanations and arguments for all design decisions and theory.
+ The literature review and framing is good.

weaknesses
- An explicit contributions statement at the end of the introduction would strengthen the paper.
- The paper could highlight the significance and benefits of the "simplification" more, e.g., what is the impact on runtime performance at deployment.

**Summary Of Recommendation:**

The paper introduces an approach and ideas with significant potential impact on robot deployment. It is well written and technically sound. The results are shown empirically on small (didactic) and real-world datasets.

---

> ### Author Response · Authors · 2022-08-23
> **Response to Reviewer UeEy**
>
> Thank you for taking the time to review our paper. We are also greatly motivated by the positive feedback and feel that our approach is well understood. We especially thank you for catching the mistake in equation (7). We will address the questions and concerns raised below. **We have also made further points in the general response to all the reviewers in a separate thread. Please see "General Response 2/2" for the updated paper and the supplementary materials.**
>
>
> > An explicit contributions statement at the end of the introduction would strengthen the paper.
>
> Thank you for your feedback. We added a list of explicit contributions at the end of Section 1.
>
> > The paper could highlight the significance and benefits of the "simplification" more, e.g., what is the impact on runtime performance at deployment.
>
> We added sample complexity analysis for our prediction-planning pipeline in Supplementary C1.2. It is confirmed that the runtime for the online planner is linear in the number of samples drawn from the predictor, and roughly 1 sec with 2000 samples (Figure 8). Based on this result, conventional risk-sensitive planning requiring many samples may have difficulties achieving real-time performance, which shows a clear benefit of our approach regarding the runtime.
>
> > In Sec. 5.2, what is $s$? Should $s = \sigma$ or is $s \geq 1$? Moreover, the penalty $\rho(x)$ is discontinuous in $x$. How does it impact performance? Also, is the problem well posed? Would $s \cdot (1 + \log(x))$ work instead of $\log(sx)$?
>
> In equation (7), s is a scaling positive value. It is a hyperparameter of the training phase. We corrected the equation to match what we had done in our experiment. The condition was changed from “x < 1” to “s*x < 1” which keeps the gradient continuous. The effect of the choice of value for s is discussed in our response to reviewer zkWD.
>
> We thank you again for your careful review. Please let us know if this addresses all of your concerns.

---

### Author Response · Authors · 2022-08-23
**General Response (1/2)**

**Comment:**

(08/24: We added the revised paper and the supplementary to General Response 1/2 as well, for extra clarity. The files are the same as the ones attached to General Response 2/2)

We sincerely thank all the reviewers and the area chair for their time and thoughtful feedback on our work. We deeply appreciate that the novelty and significance of the proposed risk-biased trajectory forecasting has been acknowledged by each reviewer and the area chair, including the proposed mathematical concepts, design decisions, experiments, and benefits such as reduced sample complexity and robustness.

In this general response, we would like to make a few remarks to respond to common concerns raised in the initial reviews, and summarize changes that we have made to the paper so far. More specific answers to each reviewer will be posted in the individual threads.

First, some of the reviewers have expressed a limited number of agents being considered for our risk-biased prediction. In fact, sophisticated multi-agent scenarios are considered in our Waymo open motion dataset (WOMD) application. Specifically, while only pairwise risk between the ego and one other agent is accounted for, the context used to make the forecast includes other agents and map information. This means that our method is able to handle marginal biased predictions for each agent in the scene, but indeed not jointly-biased predictions of the whole scene. Using marginal distributions would probably lead to overly pessimistic predictions, as potential risk-avoiding interactions would be neglected. Biasing the joint prediction would be an extension of our work and could be done with our framework, for instance by using a multi-agent cost function and a joint generative forecasting model.

Another common concern was about the generalizability of our method. While we do not observe the predictive model itself to accurately extrapolate its predictions to out-of-distribution (OOD) scenes, it does generalize to in-distribution unseen scenarios as shown by the good validation results in both the didactic and the WOMD experiments. Moreover, **we have additionally performed a prediction-planning experiment with OOD pedestrian behavior in the didactic scenario, which demonstrates the distributional robustness of our approach.** This new positive result has been added to the paper, and we hope that it strengthens our claim that a robot leveraging the RAP framework will possess distributional robustness to OOD unseen scenarios.

For issues with our references, **we have extensively checked the references again and revised any ambiguous ones that we were able to identify. We have also added a paragraph describing similarities and differences from importance sampling for rare-event simulation, in which we referenced the papers suggested by reviewer zkWD.** If any specific concerns and suggestions about the references remain, we would greatly appreciate further clarification.

**We augmented the paper and the supplementary material with sample-complexity analysis, both theoretically and empirically. We have also added information on training time and complexity for our architecture.** We hope this clarifies the benefits of the sample-complexity reduction via risk-biased prediction.

Not accounting for the ego's future path is a simplification that we made only in our WOMD experiment and that could be addressed in future work. While this might have introduced the failure cases that we presented in Fig. 4(d), we believe that it is due to a specific implementation in the WOMD experiment, not because of a simplification of our general approach. For other potential failure cases, we admit that our work does not specifically address known failures such as freezing robots or extremely low probability events. The risk-sensitivity level is a user-specified input, and if it is too high the robot might freeze due to extreme pessimism. While our work is not addressing these issues immediately, there might be interesting future research to be done by exploring connections between these challenges and risk-level specifications in the framework that we propose.

Lastly, some of the reviewers asked clarifications on the volume preservation assumption to guarantee uniqueness of the solution. The statement in the original manuscript (i.e., lines 154 – 157) was only a side note that could be misleading. Therefore **we removed the unnecessary reference to volume preservation and rephrased the corresponding sentence as simply a clue along the unsolved question about uniqueness of a solution to our optimization problem.**

(Continued to General Response 2/2)


**Zip File:**

/attachment/8a0b9a54dcd2d228ac45adc0154feb98648ff6b2.zip

---

> ### Author Response · Authors · 2022-08-23
> **General Response (2/2)**
>
> **Comment:**
>
> (Continued from General Response 1/2)
>
> **Specifically, we have made the following changes to our paper (attached)**:
> - We added an explicit contributions statement (Section 1)
> - We extensively checked the references within our paper and revised any ambiguous ones that we were able to identify.
> - Similarities and differences from importance sampling literature are clarified in the paper, along with suggested references. Unnecessary statements about volume preservation have also been revised. (Section 4)
> - We corrected the error in the equation defining the risk loss (eq. (7) in the updated paper) that reviewer UeEY pointed out.
> - In the didactic scenario, we performed a new prediction-planning experiment with OOD pedestrian behavior, and demonstrated that the proposed RAP framework is robust to distribution shifts at test time. (Section 6.3)
> - We clarified the different inputs, (ego, agent to predict, other agents and map) used in the Waymo open motion dataset experiment. (Section 6.4)
> - We completed the limitation section to reflect on a simplifying assumption of our underlying predictor model. (Section 7)
> - Complexity analyses are provided in the supplementary material, for both offline model learning and online planning. (Supplementary B, C.1.2)
> - Trajectory plots and animations are added to the supplementary material (Supplementary C.1.2 and the zip file) as requested, to give more visual information on how risk-levels affect the output of the predictor and the resulting robot plans. The same material is available on our anonymized website as well: https://sites.google.com/view/corl-risk/home
>
> **The changes we made are highlighted in green for clarity.** We believe that the overall strengths of our work have been enhanced, through various improvements such as empirical evaluation of distributional robustness and intuitive visualization of risk-biased forecasting.
> Thanks to your thorough and constructive reviews. **Please let us know if this addresses all of your remarks and questions.**
>
>
>
>
> **Zip File:**
>
> /attachment/ce464b114264d82489deb0ba3d3bcfd6f7587d41.zip

---

### Meta-Review · Area_Chair_3fr6 · 2022-08-11

**Recommendation:** Accept (Oral)
**Confidence:** 5

**Metareview:**

Summary:
This work builds a risk-aware prediction model. To construct risk-biased trajectories, it learns an additional encoder model of the latent random distribution by minimizing a cost function biased by the latent random variable.   This avoids the need to update planners when new predictors and risk measurements are evaluated and simplifies planning utilizing expectations over biased (risk-aware) distributions. The paper uses didactic and real-world datasets to explain the approach.  This novel prediction objective learns risk-biased agent trajectories. This strategy decreases online sample complexity for real-time implementation.  Experiments on a pedestrian proof-of-concept scenario and the Waymo open dataset are presented and shows an improved performance over unbiased CVAE.

Strengths
1.	The concept of developing a risk-biased trajectory forecasting framework is intriguing and novel. This forecasting technique enables the planner to be distributionally robust, that is, resilient to unmodeled sources of uncertainty.
2.	The proposed method does not necessitate any changes to the planner, and it appears that it can be integrated with a variety of planners (e.g., risk-aware planners and risk-neural planners).
3.	To forecast future states of agents, a biased distribution is constructed from which samples are drawn. With only a few samples, this biased distribution allows for accurate prediction of future states.
4.	The paper is simple to read, with strong mathematical design and extensive existence proofs accompanying each of the measurement formulae.
5.	The paper's contribution is a unique synthesis of Variational Autoencoder methods, Cross-Entropy importance sampling, and risk sensitive measures. This combination is a valuable tool for enhancing the predictive layers of perceptual-control pipelines.
6.	 The results and figures provided do a good job of demonstrating the algorithm's intuitive operation and how it effectively biases predictive distributions towards risky parts of the space.


Weaknesses
1.	A limited number of agents is considered – More sophisticated, multi-agent scenarios should be considered and a discussion on how this will scale to more agents should be in the paper.
2.	It is not clear how to handle robots failing due to the simplified prediction approach.
3.	A discussion on how generalizable the method is was not done.
4.	The citations need to be reviewed and updated due to the numerous sections in the paper requiring references.
5.	It is not clear what the usefulness of the volume preserving decoder is.
6.	Reviewers have listed some related work that has not been discussed.


**Best Paper Nomination:**

No